# Characteristics and Correlation of the Microbial Communities and Flavor Compounds during the First Three Rounds of Fermentation in Chinese Sauce-Flavor Baijiu

**DOI:** 10.3390/foods12010207

**Published:** 2023-01-03

**Authors:** Youqiang Xu, Mengqin Wu, Jialiang Niu, Mengwei Lin, Hua Zhu, Kun Wang, Xiuting Li, Baoguo Sun

**Affiliations:** 1Key Laboratory of Brewing Microbiology and Enzymatic Molecular Engineering, China General Chamber of Commerce, Beijing Technology and Business University, Beijing 100048, China; 2Beijing Advanced Innovation Center for Food Nutrition and Human Health, Beijing Technology and Business University, Beijing 100048, China; 3Beijing Huadu Wine Food Limited Liability Company, Beijing 102212, China

**Keywords:** sauce-flavor Baijiu, microbial community, flavor ester, tetramethylpyrazine, degrading metabolic pathway

## Abstract

Sauce-flavor Baijiu is representative of solid-state fermented Baijiu. It is significant to deeply reveal the dynamic changes of microorganisms in the manufacturing process and their impact on the formation of flavor chemicals correlated with the quality of Baijiu. Sauce-flavor Baijiu manufacturing process can be divided into seven rounds, from which seven kinds of base Baijius are produced. The quality of base Baijiu in the third round is significantly better than that in the first and second rounds, but the mystery behind the phenomenon has not yet been revealed. Based on high-throughput sequencing and flavor analysis of fermented grains, and correlation analysis, the concentrations of flavor chemicals in the third round of fermented grains were enhanced, including esters hexanoic acid, ethyl ester; octanoic acid, ethyl ester; decanoic acid, ethyl ester; dodecanoic acid, ethyl ester; phenylacetic acid, ethyl ester; 3-(methylthio)-propionic acid ethyl ester; acetic acid, phenylethyl ester; hexanoic acid, butyl ester, and other flavor chemicals closely related to the flavor of sauce-flavor Baijiu, such as tetramethylpyrazine. The changes in flavor chemicals should be an important reason for the quality improvement of the third round of base Baijiu. Correlation analysis showed that ester synthesis was promoted by the bacteria genus *Lactobacillus* and many low abundances of fungal genera, and these low abundances of fungal genera also had important contributions to the production of tetramethylpyrazine. Meanwhile, the degrading metabolic pathway of tetramethylpyrazine was investigated, and the possible microorganisms were correlated. These results clarified the base Baijiu quality improvement of the third round and helped to provide a theoretical basis for improving base Baijiu quality.

## 1. Introduction

Baijiu is one of the national alcoholic beverages with a history of thousands of years and one of the six world-famous distilled beverages with alcohol contents from 35–60% [1]. In 2020, the production of Baijiu reached 7.4 million kiloliters, and the sales amount exceeded 580 billion yuan. Thus Baijiu is usually recognized as an important component of the Chinese food industry. The manufacturing of Chinese Baijiu mainly includes a microbial fermentation system involving hundreds of species [1]. Baijiu brewing uses an open fermentation system, and microbes from the environment, operators, tools, water, raw materials, pits, and Daqu all participate in the fermentation process [2]. Due to the divergence of microbial ecology and varied manufacturing processes, the microbial compositions and the flavors of Baijiu products significantly diverge in different regions of China [3]. According to flavors, Chinese Baijiu is mainly divided into 12 kinds, including strong-flavor Baijiu, sauce-flavor Baijiu, light-flavor Baijiu, rice-flavor Baijiu, sesame-flavor Baijiu, laobaigan-flavor Baijiu, fuyu-flavor Baijiu, herbal-flavor Baijiu, chi-flavor Baijiu, te-flavor Baijiu, mixed-flavor Baijiu and feng-flavor Baijiu [4]. Comparing all the Baijiu manufacturing processes, sauce-flavor Baijiu shows the most complex manufacturing process and takes the longest time (one whole year) [5]. The whole manufacturing process is usually divided into seven rounds according to the production of base Baijiu (the Baijiu product before aging and blending) [4].

Of the seven rounds of grain fermentation, the first round contains the most complex steps, including two fermentation processes (Figure 1). The raw materials are steamed after moisture, cooled to room temperature, and then Daqu powder is added (the fermentation starter of Chinese Baijiu). After a short time of the accumulation process (3–5 days), the mixed materials are transferred into a fermentation container (a pit usually called Jiaochi in Chinese) for anaerobic solid-state fermentation and covered with mud (round 1a, also called Xiasha in Chinese) [5,6]. Thereafter, fermented grains are mixed with fresh raw materials, and the manufacturing process of round 1a is repeated and transferred into the Jiaochi for the second fermentation in round 1 (round 1b, also called Caosha in Chinese). Fermented grains are steamed to produce the base Baijiu product (base Baijiu 1) (Figure 1) [4]. Thereafter, the fermented grains are cooled to room temperature and added with Daqu powder. After accumulation, the mixture is transferred into the Jiaochi for fermentation for about 30 days and then steamed to obtain the Baijiu product (base Baijiu 2) [6]. In the following process, no fresh raw materials are added to the fermented grains, and all the manufacturing steps are the same as in the second round (Figure 1). Through the cyclic fermentation process, the raw materials in the fermented grains are gradually converted into seven kinds of base Baijius [4].

The sensory evaluation shows that there are divergent flavors of the seven-base Baijius. The base Baijiu from the first and second rounds shows a light flavor and without the typical sauce flavor [3,7]. The base Baijius produced by the third, fourth, and fifth rounds are generally recognized as high quality that is rich in sauce flavor, and the following sixth and seventh base Baijius decline in quality [7]. From the perspective of the manufacturing process, the second to seventh rounds of sauce-flavor Baijiu manufacturing processes are identical. Why is the quality of base Baijiu produced in round two worse than that in round three [1,3,5]? Revealing the internal causes will help to improve the manufacturing process so as to control the quality of base Baijius [3].

Studies indicate that the flavor of Baijiu is closely related to flavor chemicals produced by microbial metabolism during the manufacturing process [8]. Microbial metabolism is considered the internal driving force for the conversion of raw materials to flavor chemicals in Baijiu [2]. However, there is no systematic analysis of the relationship between microorganisms and flavor chemicals during rounds two and three, as well as the impact on the improved quality of base Baijiu three during the fermentation process of sauce-flavor Baijiu (Figure 1). Therefore, this study systematically analyzed the dynamic composition of microorganisms and flavor chemicals in the first three rounds for producing base Baijius, and the correlation between them was analyzed. The microbial metabolic mechanisms of the representative key flavor chemicals were proposed.

## 2. Materials and Methods

### 2.1. Sample Collection

Samples were obtained from a sauce-flavor Baijiu-producing company in Beijing city, China. The sorghum was the locally produced medium-size glutinous sorghum. The water content in the fermented grains was about 53–57%. Before fermentation, the steaming time of sorghum was about 1 h to ensure that the fermented grains were fully steamed without the half-cooked part inside the sorghum grain. High-temperature Daqu used for fermentation was produced by the company itself. Daqu was stored for 4–6 months before use. Daqu was first ground into powder and then added to the fermented grains, mixed and accumulated as a pile for about 5 days, and finally transferred into the Jiaochi for solid-state fermentation (Figure 1). The grain fermentation usually takes about one month [5], and samples were collected from the fermented grains in the Jiaochi about every 7 days. The samples were taken from the upper, middle, and bottom layers of the fermented grains, as shown in Appendix A. In each layer, the 5 samples were mixed together and stored at −80 °C for microbial metagenomic analysis and related flavor compound analysis. For convenience, these samples were renamed with abbreviations, as shown in Table 1.

### 2.2. Flavor Compound Analysis

The fermented grain samples were collected, and 2 g of the sample was mixed with saturated NaCl water solution, followed by ultra-sonicating for 30 min. 4-Octanol was used as the internal standard with a final concentration of 5 mg/L. DVB/CAR/PDMS fiber (50/30 μm, Supelco, Bellefonte, PA, USA) was used to extract the volatile chemicals of the sample. Volatile chemicals were analyzed using a Thermo Fisher headspace solid-phase microextraction gas chromatography-mass spectrometer (HS-SPME-GC-MS) TSQ 8000 Evo instrument (Thermo Fisher Scientific, Waltham, MA, USA) equipped with a flame ionization detector and a DB-WAX column (60 m × 0.25 mm × 0.25 μm, J&W Scientific, Folsom, CA, USA). The column temperature was maintained at 40 °C for 5 min, and then increased to 100 °C at a rate of 5 °C/min and kept for 10 min, followed by increasing to 150 °C at a rate of 5 °C/min and kept for 10 min, and finally increased to 250 °C with a rate of 5 °C/min and kept for 15 min. The helium carrier gas was used with a flow rate of 1 mL/min. Mass spectrometry (MS) was generated with an electron impact of 70 eV ionization energy and a scan range from 30 to 400 amu. Flavor chemicals were identified by matching the MS spectrum to the NIST05 database.

### 2.3. Microbial Composition Analysis

Microbial community genomic DNA was extracted using the E.Z.N.A.^®^ soil DNA Kit (Omega Bio-Tek, Norcross, GA, USA) according to the protocol. The fungal 18S rRNA genes were amplified through polymerase chain reaction (PCR) using the primer pair ITS1 (5′-TCCGTAGGTGAACCTGCGG-3′) and ITS2 (5′-GCTGCGTTCTTCATCGATGC-3′). The bacterial 16S rRNA genes were amplified through PCR with the primer pair 338F (5′-ACTCCTACGGGAGGCAGCAG-3′) and 806R (5′-GGACTACHVGGGTWTCTAAT-3′). The PCR products were prepared according to the standard protocols and sequenced using an Illumina MiSeq PE300 platform/NovaSeq PE250 platform (Illumina, San Diego, CA, USA). After quality control of reads, sequencing reads were merged by FLASH version 1.2.7 [9]. Operational Taxonomic Units (OTUs) were clustered using UPARSE version 7.1 with a 97% similarity cutoff value [10]. The taxonomy of the OTU representative sequence was analyzed by blasting the 16S rRNA and 18S rRNA gene databases using the threshold value of 70% [11]. Microbial communities were investigated based on the relative OTUs abundance at the genus level.

### 2.4. Correlation Analysis

Principal Coordinates Analysis (PCoA) was used to analyze the correlations of samples based on the microbial composition to reveal the relationship of the fermented grain samples [12]. PCoA method can reveal the simple structure hidden behind the complex data based on the selected Euclidean distance matrix. Partial Least Squares Discriminant Analysis (PLS-DA) was a multivariate statistical analysis method for discriminant analysis [13]. It judged how objects were classified according to the observed values of several variables. By microbial community analysis of the fermented grain samples, the relationships between samples could be revealed. Microbial relationship analysis was performed of the 3 dominant bacterial genera to the left bacterial genera or the fungal genera, and also the 3 dominant fungal genera to the left fungal genera or the bacterial genera using Heatmap analysis. Microbial composition and flavor chemical relationship analysis were investigated and visualized by Heatmap analysis. The Heatmap analysis calculated the correlation coefficient (Spearman rank correlation coefficient) between two groups of data and visually displayed the numerical matrix obtained through the Heatmap diagram. The data information in the two-dimensional matrix was reflected by the color change, and the color depth represented the size of the data value. The calculation and analysis of heat map were completed by R software (version 3.3.1) (R Core Team, Sydney, Australia), which was the language and operating environment for statistical analysis, mainly used for statistical calculation and mapping. The metabolic pathways were proposed based on the KEGG metabolic pathways, and the annotated genetic information of the microorganisms from Baijiu together with the literature reported.

### 2.5. Data Analysis

SPSS software (version 26.0) was used for statistical analysis. MEGA 5.2 software was used to produce the phylogenetic tree with a Neighbor-Joining method and a bootstrap value of 1000 [14].

## 3. Results and Discussion

### 3.1. Flavor Analysis of the Samples in the First Three Rounds of Sauce-Flavor Baijiu

A total of 99 flavor chemicals were detected by GC-MS, including 53 esters, 16 alcohols, 2 pyrazines, 1 furan, 8 phenols, 5 ketones, 8 aldehydes, 3 benzene, 1 ether, 1 pyrrole, and 1 thiazole in all the samples of the three rounds (Appendix A, and Figure 2). The comparative analysis of flavor chemicals in the grain samples at the end of each round showed that the concentrations of 11 flavor chemicals in the third round of fermented grains increased, including hexanoic acid, ethyl ester; octanoic acid, ethyl ester; decanoic acid, ethyl ester; dodecanoic acid, ethyl ester; phenylacetic acid, ethyl ester; 3-(methylthio)-propionic acid ethyl ester; acetic acid, phenylethyl ester; hexanoic acid, butyl ester; benzaldehyde, 2,3-butanediol, and tetramethylpyrazine. These chemicals have an important contribution to fruity, floral, and sweet flavors, as hexanoic acid, ethyl ester, showed floral and sweet flavors with an odor threshold value of 55 μg/L, and octanoic acid, ethyl ester showed a fruity flavor with an odor threshold value of 13 μg/L [2]. Decanoic acid, ethyl ester, and dodecanoic acid, ethyl ester, and other medium and long chain fatty acid esters contributed to the typical characteristics of sauce-flavor Baijiu, namely “a lingering aroma in empty glass” [15]. In addition, other important flavor chemicals, such as tetramethylpyrazine (with baking aroma and a sweet flavor), also contributed significantly to the flavor of the sauce-flavor Baijiu [16], and its concentration was significantly increased in round 3 of fermented grains. 2,3-Butanediol can provide the Baijiu product with a mellow feeling orally with a sticky and slightly sweet taste [17]. This served as the reason for the improved quality of base Baijiu in round 3.

The concentrations of two phenols decreased significantly in the fermented grains of round 3, which were 4-ethylphenol and 2-methoxy-4-ethyl-phenol (4-ethyl guaiacol, with clove and spicy flavor) [18]. 4-Ethyl guaiacol, a controversial chemical for baijiu flavor, was frequently detected in sauce-flavor Baijius in the Guizhou region of China [19,20]. However, although it was detected in the fermented grains of round 2, it was not identified in the fermented grains at the end of round 1b and round 3. This indicated the divergences in flavors between sauce-flavor Baijiu produced in the southern and northern regions of China [3,21]. The concentration of benzaldehyde (with a bitter almond flavor) [22] in fermented grains at the end of round 3 was also higher than that in round 2. A moderate amount of benzaldehyde can enrich the flavor of sauce-flavor Baijiu but also bring bitterness in taste [23]. Flavor chemicals in each fermentation round showed fluctuated concentrations, indicating the coexistence of synthesis and degradation metabolisms of these chemicals by microorganisms in the fermentation process (Figure 2). However, what were the dynamic microbial compositions during the fermentation process, and what kinds of microorganisms were related to the synthesis or degradation of the key flavor chemicals? These questions were investigated in the following experiments.

### 3.2. Microbial Community Analysis

#### 3.2.1. The Dynamic Composition of Microbial Communities during Fermentation

Venn charts can be used to analyze the common and unique microorganisms based on Operational Taxonomical Units (OTUs) in multiple groups or samples and can show the similarity and divergence of microbial compositions in samples [24]. Results showed that bacteria from the CS (Caosha, round 1b) samples included 229 microbial genera, bacteria from the C2 (round 2) sample included 235 microbial genera, and bacteria from the C3 (round 3) sample included 205 microbial genera (Figure 3a). These samples shared 160 bacterial genera, and CS samples contained 23 unique bacterial genera, C2 samples included 25 unique bacterial genera, and C3 samples contained 15 unique bacterial microbial genera. The diversity of fungi was slightly lower than that of bacteria (Figure 3b). CS samples included 141 fungal genera, C2 samples contained 169 fungal genera, and C3 samples included 209 fungal genera. These three groups of samples shared 106 genera, while CS samples contained 7 unique fungal genera, C2 samples included 22 unique fungal genera, and C3 samples contained 60 unique fungal genera (Figure 3b).

Bacterial composition analysis indicated that the dominant bacterial genera were mainly *Lactobacillus*, *Oceanobacillus,* and *Virgibacillus*. The high abundance bacterial genera in the upper, middle, and bottom layers of fermented grains were *Pediococcus*, *Virginacillus,* and *Oceanobillus* at the beginning of fermentation in round 1b (Figure 4a). During fermentation, the abundance of these three genera gradually decreased, and the abundance of *Lactobacillus* significantly increased and continued to the end of fermentation. The amount of *Lactobacillus* accounted for more than 60% of the total bacteria in the middle and late stages of fermentation. At the beginning of fermentation in round 2, the main microorganisms in the upper, middle, and bottom layers of the fermented grains were *Oceanobacillus*, *Virgibacillus,* and *Kroppenstedtia*. During fermentation, the abundance of these three genera gradually decreased, while that of *Lactobacillus* increased and showed high abundance in the middle and bottom layers of fermented grains in the middle stage and the upper, middle, and bottom layers of fermented grains in the late stage of fermentation (Figure 4a). Compared with round 1b, the increased ratio of *Lactobacillus* in round 2 was relatively slow. Compared with rounds 1b and 2, at the beginning of round 3, the abundance of *Lactobacillus* dominated in the upper, middle, and bottom layers of the fermented grains (both exceeded 57.22%), followed by *Oceanobacillus* and *Virgibacillus*, whose abundance were only 3.93–11.71% and 1.06–11.22%, respectively. During fermentation, the microbial abundance of *Lactobacillus* decreased, and the abundance of *Oceanobacillus*, *Virgibacillus*, *Kroppenstedtia,* and *Rhodococcus* increased, among which the microbial abundance of *Oceanobacillus* and *Virgibacillus* increased significantly, such as in sample C3_CJZ and C3_CJX, the abundance of *Oceanobacillus* was 19.06% and 26.00%, respectively, and that of *Virgibacillus* was 15.34% and 18.62%, respectively (Figure 4a).

Fungal composition analysis indicated that the dominant genera were *Issatchenkia*, *Byssochlamys,* and *Saccharomyces*. In round 1b, the dominant fungi at the beginning of fermentation were *Byssochlamys*, *Saccharomycopsis,* and *Issatchenkia* (Figure 4b). Thereafter, *Saccharomyces* and *Issatchenkia* became dominant fungi and lasted until the end of fermentation. In round 2, *Issatchenkia*, *Byssochlamys,* and *Saccharomyces* were the dominant fungi and lasted until the end of fermentation, and in this round, the abundance of *Monascus* was also high (2.74–23.46%). At the beginning of round 3, the dominant fungi were *Issatchenkia*, *Byssochlamys*, *Saccharomyces,* and *Thermomyces* (Figure 4b). During the fermentation process, the microbial abundance fluctuated at the genus level, but the fluctuation of microbial abundance was relatively gentle compared with those in rounds 1b and 2, except that the abundance of *Issatchenkia* was relatively high in the middle and late stages of fermentation (such as C3_14S (47.49%), C3_14Z (51.18%), and C3_21S (42.29%)) (Figure 4b).

PCoA is a non-binding data dimension reduction analysis method and can be used to study the similarity or difference of microbial composition in samples [12]. There was an obvious differentiation for the bacterial community at the beginning of fermentation, with a PC1 value of 67.07% (*p* = 0.001) (Figure 5a), but in the late stage of fermentation, bacterial composition showed a tendency of convergence with PC1 value of 68.57% (*p* = 0.012) (Figure 5b). This indicated the core functional bacteria driving the fermentation system were similar and could dominate the fermentation process, although divergences of bacterial composition were found at the beginning of fermentation (Figure 5b and Appendix A). Results also indicated that at the beginning of fermentation, bacteria from other resources could also participate in the co-fermentation system and brought relatively obvious microbial intervention, and was in accordance with previous studies that bacteria from the fermentation environment such as air, the ground of workshop, operating tools, operators and the fermentation container had a great influence on the bacterial composition and abundance [2,25]. PCoA of fungi showed results different from those of bacteria at the beginning of fermentation (Figure 5c). The divergence of fungal composition and abundance in rounds 1b, 2, and 3 were not significant. The convergence tendency of fungal composition and abundance indicated fungi from Daqu had a significant contribution to the fungi in the fermentation system, while relatively few fungi from other resources joined in the co-fermentation system (Figure 5c,d and Appendix A). This was consistent with previous studies that fungi from Daqu contributed significantly to the fungi composition in the fermented grains [25].

PLS-DA is a multivariate statistical analysis method used for discriminant analysis, which shows good sample differentiation performance based on observed or measured variable values [13]. PLS-DA results of bacteria and fungi further confirmed the PCoA results. Bacterial PLS-DA results showed that the samples were located in different phase limits with clear discrimination at the beginning of fermentation, indicating that bacteria from environmental systems had a significant impact on the bacterial composition and abundance when starting fermentation (Figure 6a). During fermentation, the distance between sample points became shorter (Figure 6a), indicating that although the fermentation rounds were different, bacteria that played a major role in the fermentation process were similar and gradually became the dominant genera in abundance. Fungal PLS-DA results were also consistent with those of PCoA. At the beginning of fermentation, there was no obvious difference between rounds 2 and 3 (Figure 6b), indicating that the fungal microorganisms in Daqu had an important influence on the microbial composition [25]. During fermentation, the succession of the fungal community showed a similar convergence phenomenon to that of bacteria.

#### 3.2.2. Microbial Interactions during Fermentation

Microbial composition analysis showed that the microbial diversity of the fermented grains in the upper, middle, and bottom layers was differentiated, indicating that the solid-state fermentation of the fermented grains showed the characteristic of heterogeneity [26] and that dividing the fermented grains into different layers could better reveal the microbial composition and abundance. The succession of microbial abundance was related to environmental conditions such as temperature, oxygen concentration, and the content of nutrients such as oligosaccharides, monosaccharides, short peptides, and amino acids that served as the metabolic substrates for microorganisms [27,28]. In addition, the interaction between microorganisms also provided an important driving force for the fluctuation of microbes [29]. In this part, the microbial interaction relationships were investigated between bacteria and bacteria, bacteria and fungi, and fungi and fungi, respectively. Results showed that there was a significant interaction between bacteria; for example, *Lactobacillus* showed negative interaction with all other bacterial genera, while *Oceanobacillus* and *Virgibacillus* both showed positive interaction with most of the bacteria except for significant negative interaction with *Lactobacillus* (Figure 7a). This might be caused by the high abundance of *Lactobacillus* in most stages of the fermentation process. A high abundance of *Lactobacillus* competitively consumed a large amount of nutrients in the fermentation system, thus inhibiting the growth of other microorganisms (Figure 7a).

The relationships between the three dominant fungal genera and the left fungal genera were relatively complicated because the abundance of dominant fungi was not as high as the abundance of *Lactobacillus* in bacteria (Figure 7b). For example, *Issatchenkia* showed significant positive interaction with *Pichia*, and negative interaction with *Byssochlamys*, *Aspergillus*, *Debaryomyces,* and *Piskurozyma*, while had no significant interaction with *Saccharomyces*, one of the dominant fungi (Figure 7b). *Byssochlamys* showed positive interaction with most fungal genera, such as *Monascus*, *Thermoascus*, *Thermomyces*, *Aspergillus*, *Leiothecium*, *Rasamsonia*, unclassified_Dipodascaceae, *Penicillium*, *Wallemia*, *Trichosporon*, unclassified_Saccharomycetales, *Naganishia*, *Geotrichum*, *Debaryomyces*, *Cystofilobasidium*, *Cladosporium*, *Rhodotorula*, *Dipodascus*, unclassified_Microascaceae, *Gymnoascus*, *Piskurozyma* and *Pithoascus*, and showed negative interaction only to *Issatchenkia*, *Saccharomyces*, *Kodamaea*, *Hyphopichia,* and *Cyberlindera*. *Saccharomyces* only showed positive interaction with *Zygosaccharomyces* but had negative interaction or no significant interactions with most fungal genera (Figure 7b).

Compared with complex positive or negative interactions between bacteria and bacteria and fungi and fungi, the interactions between bacteria and fungi were relatively insignificant (Figure 7c,d). Results of 3 dominant fungal genera with bacteria indicated *Issatchenkia* showed negative interaction with the bacterial genera *Nocardiopsis* and *Amphibacillus* and no positive or negative interaction with the left bacterial genera (Figure 7c). *Byssochlamys* showed an inhibitory relationship with *Weissella* and unclassified_Lactobacillus, and positive interaction with *Amphibacillus, Bavariicoccus*, *Clostridium* sensu stricto 18, and norank_Bacillaceae. *Saccharomyces* showed negative interaction with *Rhodococcus*, unclassified_Nocardioidaceae, and positive interaction with *Pediococcus*, *Weissella*, *Strepthmyces*, *Saccharopolyspora*, norank_Pseudonocardiaceae, and unclassified_Lactobacillus (Figure 7c). Results of 3 dominant bacterial genera with fungi showed that *Lactobacillus* showed negative interaction with *Cystofilobasidium* and *Cyberlindnera* and positive interaction with *Leiothecium*, *Rasamsonia*, *Miceoascus*, *Debaryomyces*, *Apiotrichum*, *Lichtheimia*, *Cutaneotrichosporon*, *Mucor* and *Pithoascus* (Figure 7d). *Oceanobacillus* showed negative interaction with *Saccharomyces*, *Wickerhamomyces*, *Kodamaea*, *Hyphopichia*, *Candida*, *Millerozyma*, *Mucor,* and *Dekkera*, and positive interaction with *Monascus*, unclassified_Saccharomycetales, *Naganishia,* and *Cystofilobasidium*. *Virgibacillus* showed negative interaction with *Saccharomycopsis*, *Wickerhamomyces*, *Kodamaea*, *Hyphopichia*, *Candida*, *Mucor,* and *Dekkera* and positive interaction with *Monascus* and *Cystofilobasidium* (Figure 7d).

Based on the correlation analysis, it could be concluded that due to the adaptability of *Lactobacillus* to the sauce-flavor Baijiu fermentation system, its abundance increased significantly in the middle and late stages of fermentation on each round. Due to the potential nutritional competition and the significant negative correlation between *Lactobacillus* and other bacteria (Figure 7a), *Lactobacillus* became the dominant genus of bacteria and inhibited the growth of other bacterial genera. *Issatchenkia* only showed a negative correlation with a few fungal genera, and *Byssochlamys*, another fungal genus with relatively high abundance, showed positive correlations with most of the fungal genera (Figure 7b). Therefore, there was no one fungal genus dominated in a long fermentation time.

The study of microbial interaction is very important for the establishment of a complex and stable co-microbial fermentation system [30]. For the complex fermentation system with multiple microorganisms, how to effectively control the stability and achieve products with stable flavor and quality among batches is still a challenge that enterprises are facing [31]. Based on the high-throughput analysis and functional microorganism combination, studies realized the stable production of cheese, another co-microbial fermented food, and showed that the core functional microorganisms had the ability to drive the stable fermentation of the entire system [32]. However, for most of the complex microbial fermented foods, the relevant studies are still relatively simple. How to excavate and determine the core functional microorganisms of Baijiu were proposed in previous studies with standards [2]. However, how these core microorganisms achieve the succession of microbial composition and abundance through interaction, as well as synthesize specific types of flavor substances with stable contents, still needs to be well studied. The clarity of relevant problems will help to promote the stability and quality of fermented food and is also the key to solving the problem of batch instability in the production of multi-microbial fermented food [31].

### 3.3. Correlation Analysis of Flavor Chemicals and Microbial Communities

During the manufacturing process of traditional fermented Baijiu, many microbes co-existed in the fermentation system, the abundances dynamically fluctuated, and the microbial metabolic pathways of flavor chemicals were dynamically changed [33]. Analyzing the relationship between microorganisms and flavor compounds was helpful to identify the potential functional microbial groups and control flavor synthesis during fermentation [34]. For the flavor chemicals related to product quality improvement in round 3, the correlation with microbes was analyzed (Figure 8a). The metabolic synthesis of hexanoic acid, ethyl ester was only positively correlated with *Lactobacillus*, while negatively correlated with many bacterial genera, such as *Virgibacillus*, *Weissella*, *Scopulibacillus* and *Staphylococcus* (Figure 8a). Octanoic acid, ethyl ester; phenylacetic acid, ethyl ester; 3-(methylthio)-propionic acid ethyl ester; acetic acid, phenylethyl ester, and dodecanoic acid, ethyl ester showed a positive correlation with *Lactobacillus*. Decanoic acid, ethyl ester, had no correlation with *Lactobacillus*, while hexanoic acid, butyl ester, showed a negative correlation with *Lactobacillus* (Figure 8a). These results indicated that *Lactobacillus* had an important contribution to synthesize esters and was an important functional microorganism in the fermentation process of sauce-flavor Baijiu. Compared with esters, other flavor chemicals that contribute to the quality, such as benzaldehyde, 2,3-butanediol, and tetramethylpyrazine, showed no significant correlation with *Lactobacillus* (Figure 8a). Benzaldehyde showed positive correlations with *Rhabdanaerobium*, *Clostridium* sensu stricto 15, *Novibacillus*, norank-Bacillaceae, *Thermoactinomyces*, unclassified-Bacillaceae, and *Oceanobacillus*. 2,3-Butanediol only showed a positive correlation with unclassified_Bacillaceae. No significant positive correlation was identified between tetramethylpyrazine and all of the bacterial genera. On the contrary, some bacterial genera showed a negative correlation with the production of tetramethylpyrazine, such as *Ochrobactrum* (Figure 8a). This indicated that bacterial genera played a positive role in synthesizing flavor chemicals, but for tetramethylpyrazine, an important flavor chemical in Baijiu, bacterial genera mainly showed the function of metabolic degradation (Figure 8a). This might indicate that fungi in the fermented grains promoted the metabolic synthesis of tetramethylpyrazine and investigated in the following paragraph. In addition, 4-ethyl guaiacol, usually recognized as an important flavor substance of sauce-flavor Baijiu, was positively associated with *Pedicoccus*, *Staphylococcus*, *Brevibacterium*, *Brachybacterium*, norank_Pseudonocardiaceae, and unclassified_Lactobacillales, while other bacterial genera showed negative correlation with the synthesis of 4-ethyl guaiacol, such as *Thermoactinomyces* and *Pseudomonas* (Figure 8a). Thereafter, the metabolic relationship between fungi and flavor chemicals was analyzed.

For tetramethylpyrazine, although the dominant fungal genera showed negative correlation, such as *Issatchenkia* and *Saccharomyces*, many fungal genera showed positive correlation, such as *Thermoascus*, *Thermomyces*, *Aspergillus*, *Leiothecium,* and *Rasamsonia* (Figure 8b), indicating low abundance fungi played an important role for synthesizing tetramethylpyrazine and provided a basis for regulating flavor chemical production by microbial intervention. For other important flavor chemicals in Baijiu, such as hexanoic acid, ethyl ester; octanoic acid, ethyl ester; decanoic acid, ethyl ester; dodecanoic acid, ethyl ester; phenylacetic acid, ethyl ester; 3-(methylthio)-propionic acid ethyl ester; acetic acid, phenylethyl ester; hexanoic acid, butyl ester; benzaldehyde and 2,3-butanediol, there was no significant correlation with the three dominant fungal genera *Issatchenkia*, *Byssochlamys* and *Saccharomyces* (Figure 8b), indicating that the relevant dominant fungal microorganisms had no direct contribution to the synthesis of these flavor chemicals. Microorganisms with relatively low abundance in fermented grains, such as *Thermomyces*, *Thermoascus*, *Aspergillus*, *Leiothecium,* and *Rasamsonia*, showed a positive correlation with benzaldehyde, except *Thermomyces* with no correlation (Figure 8b). Only *Saccharomyces* showed a positive correlation with 4-ethyl guaiacol, while *Cladosporium*, *Apiotrichum,* and *Leiothecium* showed a negative correlation. In addition, many low-abundance fungal genera were positively correlated with the formation of these flavor chemicals (Figure 8b). Compared with bacteria, fungi played an important role in the synthesis of important flavor chemicals in the Baijiu fermentation system. As fungi in the fermentation system are closely related to Daqu, the results provided an important basis for the importance of Daqu for Baijiu fermentation (Figure 5 and Figure 8).

### 3.4. The Synthesizing and Degrading Metabolic Pathways of Key Flavor Chemicals

Flavor chemicals fluctuated in the Baijiu fermentation process, and microorganisms contributed greatly to the synthesis of flavor chemicals (Figure 2 and Figure 8). The essence of microbial metabolism is that for a specific flavor chemical, its concentration is a dynamic balance of microbial metabolic synthesis and degradation, and microbial metabolism is a biochemical reaction catalyzed by enzymes of related metabolic pathways [35]. In order to regulate the metabolism of related flavor chemicals by microorganisms in future studies, the metabolic network of related flavor chemicals was also analyzed, especially tetramethylpyrazine. The metabolic synthesis pathways of these chemicals were summarized together with the metabolic degradation pathways. To the best of our knowledge, all previous studies only focused on the metabolic synthesis pathway of tetramethylpyrazine, and this work proposed the degradation pathway of tetramethylpyrazine in Baijiu for the first time and could provide references for the subsequent regulation of the concentration of tetramethylpyrazine in the actual fermentation process using microbes.

The ethyl ester metabolic synthesis pathway used starch and protein substances contained in fermented grains as the substrates, and starch was converted into the key metabolic intermediate pyruvate through glycolysis, pyruvate was converted to acetaldehyde by pyruvate decarboxylase, and acetaldehyde could be converted to acetic acid by aldehyde dehydrogenase, or to ethanol by alcohol dehydrogenase (Figure 9a). Pyruvate could also be converted to acetyl-CoA by 2-oxoacid oxidoreductase. The key metabolic intermediate acetyl-CoA served as the substrate for synthesizing medium and long chain fatty acids through the reverse β-oxidation pathway [36] and generated the precursors of flavor ethyl esters, such as hexanoic acid, octanoic acid, decanoic acid, and dodecanoic acid (Figure 9a). These acids were converted into the corresponding esters through an esterification reaction with alcohol (such as ethanol or butanol) (Figure 9a). Protein in grains could be hydrolyzed by a protease to produce amino acids, such as L-phenylalanine, and further converted to phenylpyruvic acid by amino acid transaminase, and finally produced phenylethanol and phenylacetic acid. Phenylethanol and phenylacetic acid could react with acetic acid and ethanol, respectively, to produce phenylacetic acid, ethyl ester, and acetic acid, phenylethyl ester (Figure 9a). Enzymes catalyzing esterification reactions by microorganisms from Baijiu had been reported in previous studies, such as carboxylesterase, and provided a theoretical basis for esterification using acids and alcohols as substrates [2]. In addition, short-chain fatty acid esters such as ethyl acetate could be produced through transesterification, which was also an important metabolic pathway to generate esters in Baijiu [37]. Meanwhile, enzymatic esterification was a reversible process, and microorganisms could also hydrolyze the ester bond through similar enzymes and then consume the acid or alcohol as the carbon source generated from ester hydrolysis [38].

Tetramethylpyrazine, as a health factor to alleviate cardiovascular and cerebrovascular diseases, has received great attention since its identification from Baijiu [39]. The metabolic synthetic pathway of tetramethylpyrazine has been revealed by microbial metabolism combined with a one-step chemical reaction (Figure 9b). The direct precursor of tetramethylpyrazine is acetoin, catalyzed by α-acetolactate decarboxylase (AlsD) using α-acetolactate as the substrate, and α-acetolactate was produced by α-acetolactate synthetase (AlsS) using pyruvic acid as the substrate (Figure 9b). Acetoin was then reacted with NH_4_^+^ produced by amino acid metabolism to synthesize tetramethylpyrazine [40]. Acetoin also served as the substrate to produce 2,3-butanediol catalyzed by 2,3-butanediol dehydrogenase (BdhA). Flavor analysis showed that the concentration of tetramethylpyrazine in the fermented grains fluctuated, indicating that tetramethylpyrazine was likely also degraded during fermentation [41]. There was only one piece of literature about microbial tetramethylpyrazine degradation. Tetramethylpyrazine was converted to (*Z*)-*N*,*N*′-(but-2-ene-2,3-diyl)diacetamide by tetramethylpyrazine oxygenase (TpdAB), and then generated *N*-(3-oxobutan-2-yl)acetamide by (*Z*)-*N*,*N*′-(but-2-ene-2,3-diyl)diacetamide hydrolase (TpdC), and further converted to *N*-(3-hydroxybutan-2-yl)acetamide by aminoalcohol dehydrogenase (TpdE) (Figure 9b). Correlation analysis between Baijiu microorganisms and tetramethylpyrazine showed that some bacterial genera could degrade tetramethylpyrazine, such as *Ochrobactrum*, *Saccharomonospora*, *Enterococcus*, *Klebsiella*, *Kocuria*, unclassified_Lactobacillales, *Pseudomonas*, *Brachybacterium* and *Streptomyces*. As scarcely any studies about Baijiu microorganisms on tetramethylpyrazine degradation and the ring opening step was crucial in degradation, we searched NCBI and PDB databases based on the sequence of TpdA, responsible for ring opening catalysis, followed by phylogenetic analysis (Appendix A). It was found that some microorganisms, such as *Ochrobactrum*, *Pseudomonas*, *Klebsiella,* and *Streptomyces*, carried relevant functional enzymes with a sequence similarity of more than 30% to the enzyme from *Rhodococcus* with known catalytic function (Appendix A), indicating the potential function for degrading tetramethylpyrazine, but needed further verification. In addition, other chemicals that played an important role in quality improvement, such as 3-(methylthio)-propionic acid ethyl ester and 4-ethyl guaiacol, the microbial synthesis and degradation of 3-(methylthio)-propionic acid ethyl ester had not been investigated [42], while the metabolic synthesis of 4-ethyl guaiacol was reported. Related microorganisms, such as *Wickerhamiella versatilis*, used ferulic acid as the substrate to produce 4-vinyl guaiacol by ferulic acid decarboxylase and then generated 4-ethyl guaiacol by 4-vinyl guaiacol reductase [43]. Up to now, the metabolic degradation of 4-ethyl guaiacol was not reported.

The metabolism of microorganisms in complex fermented systems is a hot topic in the industry and also a bottleneck. Nowadays, methods have been developed for investigating metabolic networks of microorganisms in complex fermentation systems, such as metagenome sequencing and metatranscriptome sequencing coupled with metabonomics analysis, which provide techniques for revealing real-time dynamic metabolism of flavor chemicals during fermentation [44]. All these analyses are based on the clarity of metabolic networks of flavor chemicals. The summary of the metabolic network of flavor chemicals in this work can provide a theoretical reference for the analysis of real-time dynamic metabolism of flavor chemicals in the fermentation process in future works.

## 4. Conclusions

In this work, the flavor chemicals and microbial communities of fermented grains in the first three rounds of the base Baijiu manufacturing process were analyzed and correlated. Results indicated that the increased concentrations of important chemicals such as hexanoic acid, ethyl ester; octanoic acid, ethyl ester; decanoic acid, ethyl ester; dodecanoic acid, ethyl ester; phenylacetic acid, ethyl ester; 3-(methylthio)-propionic acid, ethyl ester; acetic acid, phenylethyl ester; hexanoic acid, butyl ester, 2,3-butanediol, and tetramethylpyrazine were related to improving the quality of base Baijius. One of the dominant bacterial genera, *Lactobacillus*, and many of the fungal genera with low abundances showed positive interaction with the synthesis of esters, while many low-abundance fungal genera showed positive interaction with the synthesis of tetramethylpyrazine. This work also proposed the synthetic pathway of flavor esters and the synthesis and degradation pathways of tetramethylpyrazine. The above results provide basic data references for the quality improvement of base Baijius in different rounds of sauce-flavor Baijiu.

## Figures and Tables

**Figure 1 foods-12-00207-f001:**
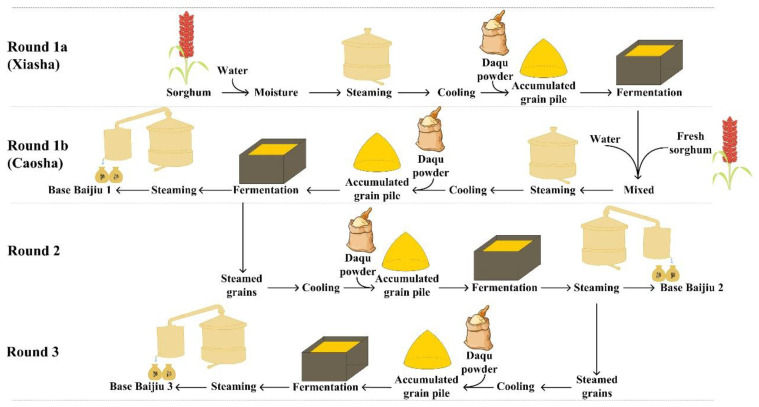
The first three rounds of the fermentation process of sauce-flavor Baijiu.

**Figure 2 foods-12-00207-f002:**
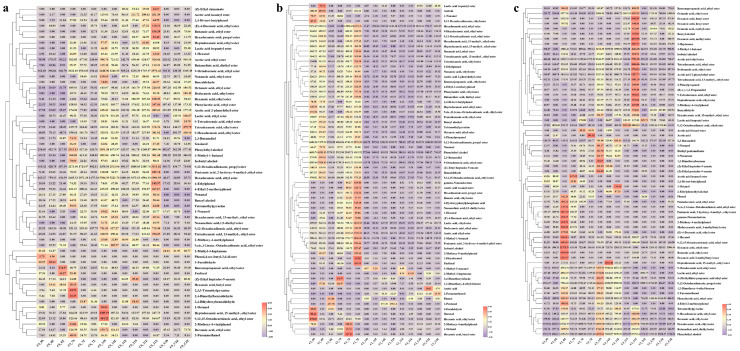
Flavor chemical analysis of the fermented grains from rounds 1b, 2, and 3 during the sauce-flavor Baijiu manufacturing process. (**a**), round 1b. (**b**), round 2. (**c**) round 3.

**Figure 3 foods-12-00207-f003:**
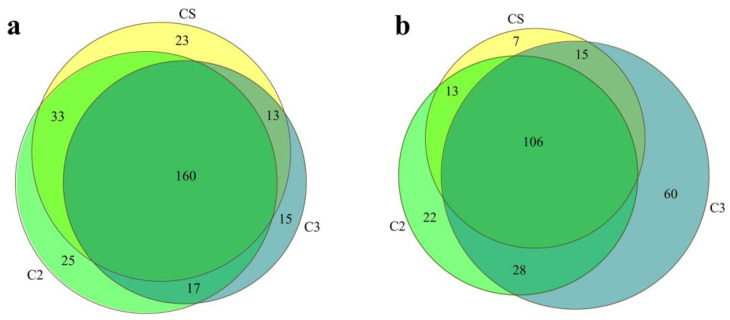
Analysis of the common and unique microorganisms from fermented grains at the genus level. (**a**) bacteria. (**b**) fungi. CS, round 1b. C2, round 2. C3, round 3.

**Figure 4 foods-12-00207-f004:**
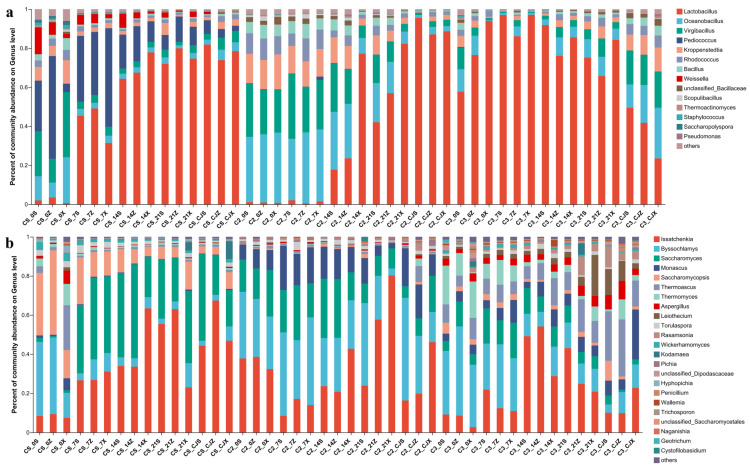
The microbial composition analysis of bacteria and fungi at genus level in rounds 1b, 2, and 3 of sauce-flavor Baijiu samples. (**a**) bacteria. (**b**) fungi.

**Figure 5 foods-12-00207-f005:**
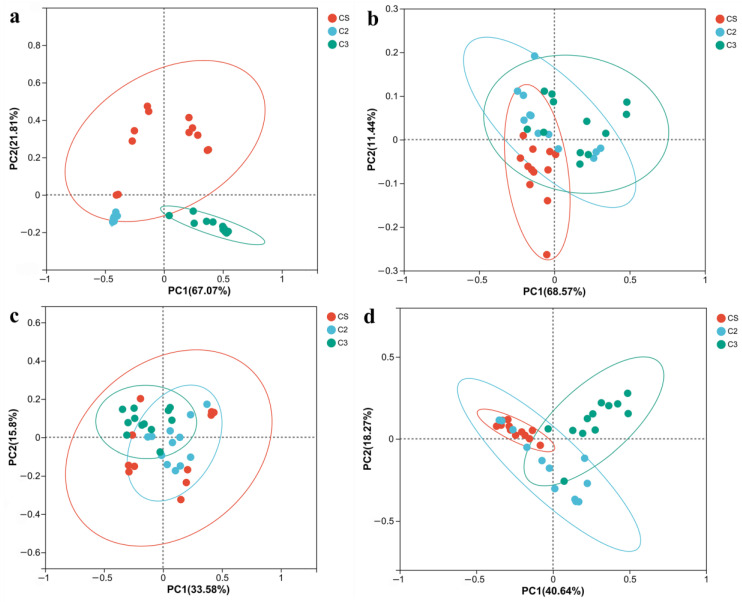
PCoA analysis of the microbial genera in the fermented grains during the sauce-flavor Baijiu manufacturing process. (**a**) bacteria from 0 and 7th d samples. (**b**) bacteria from 21st d and fermented end samples. (**c**) fungi from 0 and 7th d samples. (**d**) fungi from 21st d and fermented end samples.

**Figure 6 foods-12-00207-f006:**
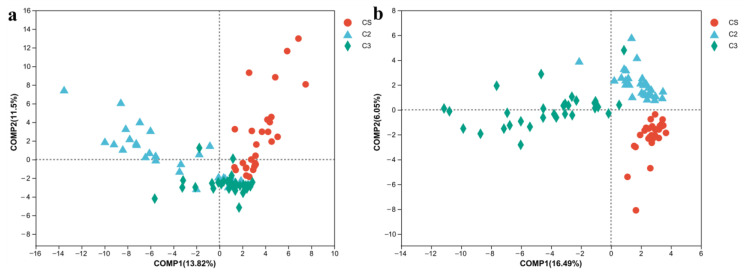
PLS-DA analysis of the microbial genera in the fermented grains during the sauce-flavor Baijiu manufacturing process. (**a**) bacteria. (**b**) fungi. CS, round 1b. C2, round 2. C3, round 3.

**Figure 7 foods-12-00207-f007:**
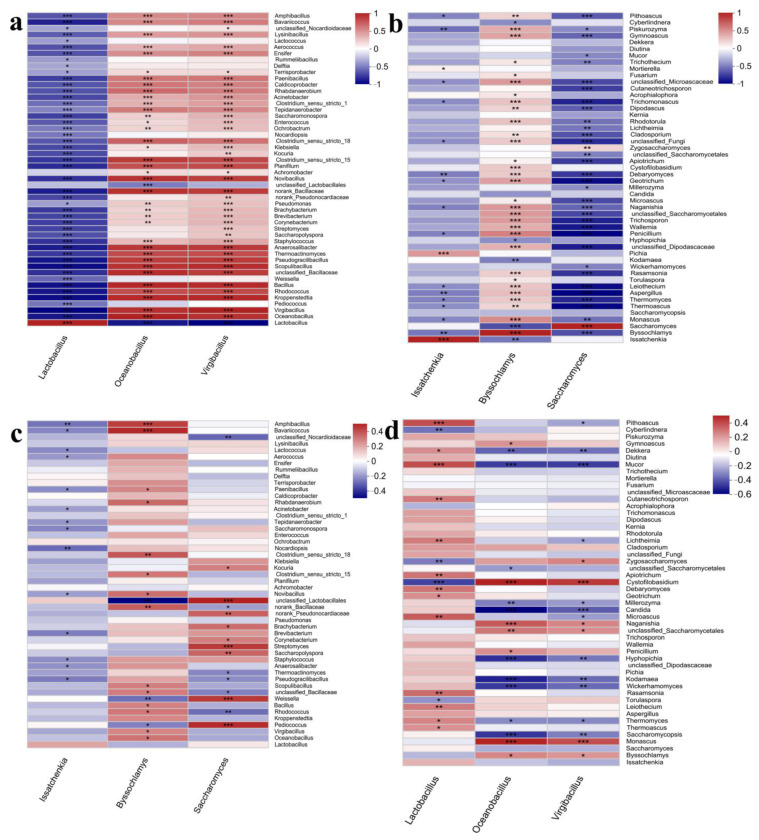
Reciprocal interaction analysis of the microorganisms in the fermented grains during the manufacturing process of sauce-flavor Baijiu. (**a**) the three dominant bacteria genera with other bacteria. (**b**) the three dominant fungi genera with other fungi. (**c**) the three dominant fungi genera with bacteria. (**d**) the three dominant bacteria genera with fungi. * 0.01 < *p* ≤ 0.05, ** 0.001 < *p* ≤ 0.01, *** *p* ≤ 0.001.

**Figure 8 foods-12-00207-f008:**
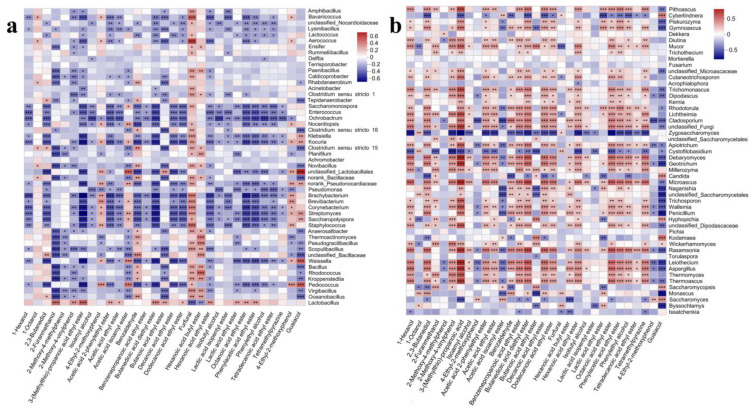
Correlation analysis of flavor chemicals and microbial community in the fermented grains of sauce-flavor Baijiu. Red represents a positive correlation, and blue represents a negative correlation. (**a**), bacteria with flavor chemicals. (**b**), fungi with flavor chemicals. * 0.01 < *p* ≤ 0.05, ** 0.001 < *p* ≤ 0.01, *** *p* ≤ 0.001.

**Figure 9 foods-12-00207-f009:**
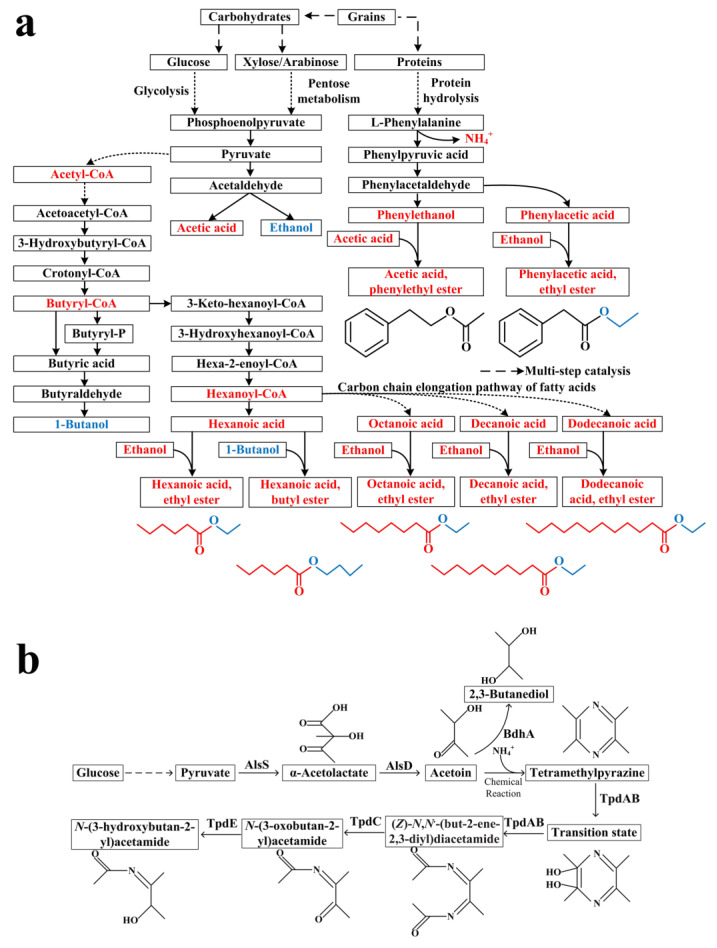
The synthetic pathway of esters (**a**) and the synthetic and degrading pathway of tetramethylpyrazine (**b**).

**Table 1 foods-12-00207-t001:** Abbreviation and detail information of samples used for sequencing.

Sample Number	The Data and Position of Samples	Sample Number for Sequencing in Duplicate
CS_0S	Round 1b, the 0 day, fermented grains in upper layer of Jiaochi	CS_0S_1, CS_0S_2
CS_0Z	Round 1b, the 0 day, fermented grains in middle layer of Jiaochi	CS_0Z_1, CS_0Z_2
CS_0X	Round 1b, the 0 day, fermented grains in bottom layer of Jiaochi	CS_0X_1, CS_0X_2
CS_7S	Round 1b, the 7th day, fermented grains in upper layer of Jiaochi	CS_7S_1, CS_7S_2
CS_7Z	Round 1b, the 7th day, fermented grains in middle layer of Jiaochi	CS_7Z_1, CS_7Z_2
CS_7X	Round 1b, the 7th day, fermented grains in bottom layer of Jiaochi	CS_7X_1, CS_7X_2
CS_14S	Round 1b, the 14th day, fermented grains in upper layer of Jiaochi	CS_14S_1, CS_14S_2
CS_14Z	Round 1b, the 14th day, fermented grains in middle layer of Jiaochi	CS_14Z_1, CS_14Z_2
CS_14X	Round 1b, the 14th day, fermented grains in bottom layer of Jiaochi	CS_14X_1, CS_14X_2
CS_21S	Round 1b, the 21st day, fermented grains in upper layer of Jiaochi	CS_21S_1, CS_21S_2
CS_21Z	Round 1b, the 21st day, fermented grains in middle layer of Jiaochi	CS_21Z_1, CS_21Z_2
CS_21X	Round 1b, the 21st day, fermented grains in bottom layer of Jiaochi	CS_21X_1, CS_21X_2
CS_CJS	Round 1b, fermented grains in up layer of Jiaochi at the end of fermentation	CS_CJS_1, CS_CJS_2
CS_CJZ	Round 1b, fermented grains in middle layer of Jiaochi at the end of fermentation	CS_CJZ_1, CS_CJZ_2
CS_CJX	Round 1b, fermented grains in bottom layer of Jiaochi at the end of fermentation	CS_CJX_1, CS_CJX_2
C2_0S	Round 2, the 0 day, fermented grains in up layer of Jiaochi	C2_0S_1, C2_0S_2
C2_0Z	Round 2, the 0 day, fermented grains in middle layer of Jiaochi	C2_0Z_1, C2_0Z_2
C2_0X	Round 2, the 0 day, fermented grains in bottom layer of Jiaochi	C2_0X_1, C2_0X_2
C2_7S	Round 2, the 7th day, fermented grains in up layer of Jiaochi	C2_7S_1, C2_7S_2
C2_7Z	Round 2, the 7th day, fermented grains in middle layer of Jiaochi	C2_7Z_1, C2_7Z_2
C2_7X	Round 2, the 7th day, fermented grains in bottom layer of Jiaochi	C2_7X_1, C2_7X_2
C2_14S	Round 2, the 14th day, fermented grains in up layer of Jiaochi	C2_14S_1, C2_14S_2
C2_14Z	Round 2, the 14th day, fermented grains in middle layer of Jiaochi	C2_14Z_1, C2_14Z_2
C2_14X	Round 2, the 14th day, fermented grains in bottom layer of Jiaochi	C2_14X_1, C2_14X_2
C2_21S	Round 2, the 21st day, fermented grains in up layer of Jiaochi	C2_21S_1, C2_21S_2
C2_21Z	Round 2, the 21st day, fermented grains in middle layer of Jiaochi	C2_21Z_1, C2_21Z_2
C2_21X	Round 2, the 21st day, fermented grains in bottom layer of Jiaochi	C2_21X_1, C2_21X_2
C2_CJS	Round 2, fermented grains in up layer of Jiaochi at the end of fermentation	C2_CJS_1, C2_CJS_2
C2_CJZ	Round 2, fermented grains in middle layer of Jiaochi at the end of fermentation	C2_CJZ_1, C2_CJZ_2
C2_CJX	Round 2, fermented grains in bottom layer of Jiaochi at the end of fermentation	C2_CJX_1, C2_CJX_2
C3_0S	Round 3, the 0 day, fermented grains in up layer of Jiaochi	C3_0S_1, C3_0S_2
C3_0Z	Round 3, the 0 day, fermented grains in middle layer of Jiaochi	C3_0Z_1, C3_0Z_2
C3_0X	Round 3, the 0 day, fermented grains in bottom layer of Jiaochi	C3_0X_1, C3_0X_2
C3_7S	Round 3, the 7th day, fermented grains in up layer of Jiaochi	C3_7S_1, C3_7S_2
C3_7Z	Round 3, the 7th day, fermented grains in middle layer of Jiaochi	C3_7Z_1, C3_7Z_2
C3_7X	Round 3, the 7th day, fermented grains in bottom layer of Jiaochi	C3_7X_1, C3_7X_2
C3_14S	Round 3, the 14th day, fermented grains in up layer of Jiaochi	C3_14S_1, C3_14S_2
C3_14Z	Round 3, the 14th day, fermented grains in middle layer of Jiaochi	C3_14Z_1, C3_14Z_2
C3_14X	Round 3, the 14th day, fermented grains in bottom layer of Jiaochi	C3_14X_1, C3_14X_2
C3_21S	Round 3, the 21st day, fermented grains in up layer of Jiaochi	C3_21S_1, C3_21S_2
C3_21Z	Round 3, the 21st day, fermented grains in middle layer of Jiaochi	C3_21Z_1, C3_21Z_2
C3_21X	Round 3, the 21st day, fermented grains in bottom layer of Jiaochi	C3_21X_1, C3_21X_2
C3_CJS	Round 3, fermented grains in up layer of Jiaochi at the end of fermentation	C3_CJS_1, C3_CJS_2
C3_CJZ	Round 3, fermented grains in middle layer of Jiaochi at the end of fermentation	C3_CJZ_1, C3_CJZ_2
C3_CJX	Round 3, fermented grains in bottom layer of Jiaochi at the end of fermentation	C3_CJX_1, C3_CJX_2

## Data Availability

Data are contained within the article. All the data generated for this study are available on request to the corresponding author.

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
