# Peer review of "Characteristics and Correlation of the Microbial Communities and Flavor Compounds during the First Three Rounds of Fermentation in Chinese Sauce-Flavor Baijiu"

_foods, 2023, doi:10.3390/foods12010207_

Round 1

Reviewer 1 Report

The manuscript describes the microbial communities present during the first 3 stages of Baijiu (sauce-flavor) and later correlates this with the flavor profile. As is the information is very interesting and thoughtful. The introduction is well-supported and the materials and methods are well described. Results are presented and discussed accordingly to relevant findings.

However, I have some observations:

Introduction: A brief description of what Baijiu is should be included. Also the importance of the product in the market. 

Methodology: Please include the methods used to determine the metabolic pathways shown in the manuscript. 

Results: 

1. How do the authors conclude that Daqu did not have an influence in the bacterial communities, nor did have an influence in the fungal communities? I think that a microbial characterization of the Daqu should be included to support this. 

2. A clear conclusion on how the interactions allow the primary bacteria/fungi to become dominant should be included. 

3. Why only round 3 was analyzed for flavor-microorganisms relationships?

4. Did the authors quantify the amount of the major (important) flavor compounds? If so, please include the information indicating also its relevance to threshold levels. 

Author Response

Response to Reviewer 1 Comments point by point

The manuscript describes the microbial communities present during the first 3 stages of Baijiu (sauce-flavor) and later correlates this with the flavor profile. As is the information is very interesting and thoughtful. The introduction is well-supported and the materials and methods are well described. Results are presented and discussed accordingly to relevant findings.

However, I have some observations:

Introduction: A brief description of what Baijiu is should be included. Also the importance of the product in the market.

Response: Thanks for this suggestion. The following information was added in the revised manuscript.

Baijiu is one of the national alcoholic beverages with a history of thousands of years, and one of the six world famous distilled beverages with alcohol contents from 35% - 60% [1]. In 2020, the production of Baijiu reaches 7.4 million kiloliters and the sales amount exceeds 580 billion yuan, thus Baijiu is usually recognized as an important component of Chinese food industry.

Methodology: Please include the methods used to determine the metabolic pathways shown in the manuscript.

Response: Thanks for this suggestion. The following information was added in the revised manuscript.

The metabolic pathways were proposed based on the KEGG metabolic pathways, and the annotated genetic information of the microorganisms from Baijiu together with the literatures reported.

Results:

  1. How do the authors conclude that Daqu did not have an influence in the bacterial communities, nor did have an influence in the fungal communities? I think that a microbial characterization of the Daqu should be included to support this.

Response: Thanks for this comment.

As the fermentation starter for Baijiu, Daqu made significant contribution to Baijiu microorganisms. Daqu provided both bacteria and fungi resources for Baijiu fermentation. However, due to open fermentation operation of Baijiu manufacturing process, microorganisms from other sources besides Daqu also contributed to microorganisms for grain fermentation (Food Chem. 2022, 369, 130920; Appl Environ Microbiol. 2018, 84(4): e02369-17). For example, high-throughput sequencing combined with multiphasic metabolite target analysis was applied to study the microbial succession and metabolism changes during Chinese Baijiu fermentation. SourceTracker was applied to evaluate the contribution of environmental microbiota to fermentation. Results showed that Daqu contributed 9.10% to 27.39% of bacterial communities and 61.06% to 80.00% of fungal communities to fermentation, whereas environments (outdoor ground, indoor ground, tools, and other unknown environments) contributed 62.61% to 90.90% of bacterial communities and 20.00% to 38.94% of fungal communities to fermentation (Appl Environ Microbiol. 2018, 84(4): e02369-17). Studies indicated that Daqu contributed more to the fungi of Baijiu fermentation, while it also contributed important bacterial resources. In this work, PCoA and PLS-DA analysis showed that at the beginning of the three rounds of fermentation, the microbial compositions of bacteria were diverged, indicating that the participation of bacteria from the environment brought microbial disturbance. Compared the data of fungal composition and abundance in the three rounds of fermentation, the difference was not obvious, indicating that the environmental fungal microorganisms had less disturbance to the microbial composition of fermented grains. We revised the related discussion to make the conclusions more objective and scientific. Due to the huge amount of experiments and the time consume of high throughput sequencing and analysis, we did not carry out high-throughput sequencing and annotation of Daqu microbes in this part, as well as correlation analysis with microorganisms in fermented grains. The sentences in the manuscript were revised as follows.

PCoA is a non-binding data dimension reduction analysis method, and can be used to study the similarity or difference of microbial composition in samples [12]. There was an obvious differentiation for bacterial community at the beginning of fermentation, with PC1 value of 67.07% (p = 0.001) (Fig. 5a), but in the late stage of fermentation, bacterial composition showed a tendency of convergence with PC1 value of 68.57% (p = 0.012) (Fig. 5b). This indicated the core functional bacteria driving the fermentation system were similar and could dominate the fermentation process, although divergences of bacterial composition were found at the beginning of fermentation (Fig. 5b and S2a). Results also indicated that at the beginning of fermentation, bacteria from other resources could also participate into the co-fermentation system and brought relatively obvious microbial intervention, and was in accordance with previous studies that bacteria from the fermentation environment such as air, the ground of workshop, operating tools, operators and the fermentation container had a great influence on the bacterial composition and abundance [2, 25]. PCoA of fungi showed results different from those of bacteria at the beginning of fermentation (Fig. 5c). The divergence of fungal composition and abundance in rounds 1b, 2 and 3 were not significant. The convergence tendency of fungal composition and abundance indicated fungi from Daqu had a significant contribution to the fungi in the fermentation system while relatively few fungi from other resources joined in the co-fermentation system (Fig. 5c, 5d and S2b). This was consistent with previous studies that fungi from Daqu contributed significantly to the fungi composition in the fermented grains [25].

  1. A clear conclusion on how the interactions allow the primary bacteria/fungi to become dominant should be included.

Response: Thanks for this suggestion. The following discussion was added in the revised manuscript.

Based on the correlation analysis, it could be concluded that due to the adaptability of Lactobacillus to the sauce-flavor Baijiu fermentation system, its abundance increased significantly in the middle and late stages of fermentation on each round. Due to the potential nutritional competition and the significant negative correlation between Lactobacillus and other bacteria (Fig. 7a), Lactobacillus became the dominant genus of bacteria, and inhibited the growth of other bacterial genera. Issatchenkia only showed negative correlation with few fungal genera, and Byssochlamys, another fungal genus with relative high abundance, showed positive correlations with most of fungal genera (Fig. 7b). Therefore, there was no one fungal genus dominated in a long fermentation time.

  1. Why only round 3 was analyzed for flavor-microorganisms relationships?

Response: Thanks for this suggestion.

Based on flavor comparison analysis of round 3 with those of round 1b and round 2 (Fig. 2), the main flavor substances might improve the flavor of fermented grains in round 3 were identified. Therefore, in this part we mainly focused on the round 3 for relationship analysis between flavor chemicals and microorganisms. Thereafter, the microorganisms for flavor synthesis could be revealed through correlation analysis, so as to provide reference for the quality improvement of the manufacturing process. Analysis of the correlation between microorganisms and flavor chemicals was based on rigorous algorithms and was reliable. In the future, targeted identification of functional microorganisms and investigating fermentation characteristics could be carried out based on the combination of culturomics and microbial fermentation to further verify the correlation analysis of flavor chemicals and microorganisms. Meanwhile, scientific study is a continuous work. Considering the aim of this work, it was mainly to find microorganisms that might be related to the quality improvement of round 3 correlated with the synthesis or degradation of important flavor chemicals. This aim was achieved, thus no more other correlation analysis was performed.

  1. Did the authors quantify the amount of the major (important) flavor compounds? If so, please include the information indicating also its relevance to threshold levels.

Response: Thanks very much for the suggestion.

As a huge designed investigation of the whole manuscript, we did not put many effects on quantification of the amount of major (important) flavor compounds. We will establish methods for preciously quantifying the important flavor compounds in our future work, and based on this, the flavor contribution of these chemicals can be further evaluated by sensory evaluation such as aroma recombination, and omission studies.

We hope that the answers above will enable our manuscript to win your satisfaction.

We are looking forward to hearing further information from you and would like to do as required if necessary.

Reviewer 2 Report

The presented work is interesting from the point of view of basic research as well as it has an application character. 

Some details whcich in my opinion need to be improved :

1. There is a lack of information on the raw materials used:

type of sorghum, sorghum: water ratio, moisture in % , steaming conditions Daqu powder (producer, dosage), …………………. etc.

2. line 74 – 77: “ From the perspective of manufacturing process, the second to seventh rounds of sauce-flavor Baijiu manufacturing processes are identical. Why is the quality of base Baijiu produced in round 2 worse than that in round 3? Revealing the internal causes will help to improve the manufacturing process, so as to control the quality of base Baijius”

here references needed

3. The microbial interactions during fermentation – lack of methodology , how was it evaluated

4. There is a lot of results, the graphical representation of which is very difficult. However

Figure 2 is very difficult to read

Figure 3 I suggest you change the chart type. The size of the wheels is not suitable for the value, and it does not make sense.

Author Response

Response to Reviewer 2 Comments point by point

The presented work is interesting from the point of view of basic research as well as it has an application character.

Some details which in my opinion need to be improved:

  1. There is a lack of information on the raw materials used:

type of sorghum, sorghum: water ratio, moisture in % , steaming conditions Daqu powder (producer, dosage), …………………. etc.

Response: Thanks for the suggestion. We added the following information in the “Materials and methods” of the revised manuscript.

Samples were obtained from a sauce-flavor Baijiu producing company in Beijing city, China. The sorghum was the local produced medium-size glutinous sorghum. The water content in the fermented grains was about 53% - 57%. Before fermentation, the steaming time of sorghum was about 1 hour to ensure that the fermented grains were fully steamed without half-cooked part inside the sorghum grain. High-temperature Daqu used for fermentation was produced by the company itself. Daqu was stored for 4 – 6 months before usage. Daqu was first ground into powder, and then added to the fermented grains, mixed and accumulated as a pile for about 5 days, and finally transferred into the Jiaochi for solid state fermentation (Fig. 1).

  1. line 74 – 77: “ From the perspective of manufacturing process, the second to seventh rounds of sauce-flavor Baijiu manufacturing processes are identical. Why is the quality of base Baijiu produced in round 2 worse than that in round 3? Revealing the internal causes will help to improve the manufacturing process, so as to control the quality of base Baijius”

here references needed

Response: Thanks for the comment. References were cited as follows.

From the perspective of manufacturing process, the second to seventh rounds of sauce-flavor Baijiu manufacturing processes are identical. Why is the quality of base Baijiu produced in round 2 worse than that in round 3 [1, 3, 5]? Revealing the internal causes will help to improve the manufacturing process, so as to control the quality of base Baijius [3].

  1. The microbial interactions during fermentation – lack of methodology , how was it evaluated.

Response: Thanks for the comment. The following information was added in the revised manuscript.

Microbial relationship analysis was performed of the 3 dominant bacterial genera to the left bacterial genera or the fungal genera, and also the 3 dominant fungal genera to the left fungal genera or the bacterial genera using Heatmap analysis. Microbial composition and flavor chemical relationship analysis was investigated and visualized by Heatmap analysis. The Heatmap analysis calculated the correlation coefficient (Spearman rank correlation coefficient) between two groups of data, and visually displayed the numerical matrix obtained through the Heatmap diagram. The data information in the two-dimensional matrix was reflected by the color change, and the color depth represented the size of the data value. The calculation and analysis of heat map analysis was completed by R software (version 3.3.1), which was the language and operating environment for statistical analysis, mainly used for statistical calculation and mapping.

  1. There is a lot of results, the graphical representation of which is very difficult. However, Figure 2 is very difficult to read.

Figure 3 I suggest you change the chart type. The size of the wheels is not suitable for the value, and it does not make sense.

Response: Thanks for this suggestion.

The font in Figure 2 is truly very small, but for fermented grains, flavor analysis shows that there are indeed many flavor chemicals, which need to be presented as a whole to have an intuitive understanding of the relative content and changes of flavor chemicals during the whole fermentation process. To increase the readability of data, we provide corresponding Tables S1, S2 and S3 in the supplementary information file, which can directly offer all data information.

We revised Figure 3 as follows.

[The picture of Figure 3 could be seen in the attached file]

Figure 3.  Analysis of the common and unique microorganisms from fermented grains in genus level. (a), bacteria. (b) fungi. CS, round 1b. C2, round 2. C3, round 3.

We hope that the answers above will enable our manuscript to win your satisfaction.

We are looking forward to hearing further information from you and would like to do as required if necessary.